# The Role of Leaky Gut in Nonalcoholic Fatty Liver Disease: A Novel Therapeutic Target

**DOI:** 10.3390/ijms22158161

**Published:** 2021-07-29

**Authors:** Takaomi Kessoku, Takashi Kobayashi, Kosuke Tanaka, Atsushi Yamamoto, Kota Takahashi, Michihiro Iwaki, Anna Ozaki, Yuki Kasai, Asako Nogami, Yasushi Honda, Yuji Ogawa, Shingo Kato, Kento Imajo, Takuma Higurashi, Kunihiro Hosono, Masato Yoneda, Haruki Usuda, Koichiro Wada, Satoru Saito, Atsushi Nakajima

**Affiliations:** 1Department of Gastroenterology and Hepatology, Yokohama City University Graduate School of Medicine, 3-9 Fukuura, Kanazawa-ku, Yokohama 236-0004, Japan; tkhkcb@gmail.com (T.K.); kosuke.tsssik@gmail.com (K.T.); atsushi.y.0410@gmail.com (A.Y.); takahashi1700pk9@gmail.com (K.T.); michihir@yokohama-cu.ac.jp (M.I.); anx0513ro@hotmail.com (A.O.); y.kasai.91@gmail.com (Y.K.); nogamia@yokohama-cu.ac.jp (A.N.); yhonda@yokohama-cu.ac.jp (Y.H.); ogaway@yokohama-cu.ac.jp (Y.O.); shingo800m@hotmail.com (S.K.); kento318@yokohama-cu.ac.jp (K.I.); takuma_h@yokohama-cu.ac.jp (T.H.); hiro1017@yokohama-cu.ac.jp (K.H.); yoneda@yokohama-cu.ac.jp (M.Y.); ssai1423@yokohama-cu.ac.jp (S.S.); nakajima-tky@umin.ac.jp (A.N.); 2Department of Palliative Medicine, Yokohama City University Hospital, 3-9 Fukuura, Kanazawa-ku, Yokohama 236-0004, Japan; 3Department of Pharmacology, Shimane University Faculty of Medicine, 89-1 Enyacho, Izumo, Shimane 693-8501, Japan; h-usuda@med.shimane-u.ac.jp (H.U.); koiwada@med.shimane-u.ac.jp (K.W.)

**Keywords:** leaky gut, gut permeability, nonalcoholic fatty liver disease, nonalcoholic steatohepatitis, endotoxin

## Abstract

The liver directly accepts blood from the gut and is, therefore, exposed to intestinal bacteria. Recent studies have demonstrated a relationship between gut bacteria and nonalcoholic fatty liver disease (NAFLD). Approximately 10–20% of NAFLD patients develop nonalcoholic steatohepatitis (NASH), and endotoxins produced by Gram-negative bacilli may be involved in NAFLD pathogenesis. NAFLD hyperendotoxicemia has intestinal and hepatic factors. The intestinal factors include impaired intestinal barrier function (leaky gut syndrome) and dysbiosis due to increased abundance of ethanol-producing bacteria, which can change endogenous alcohol concentrations. The hepatic factors include hyperleptinemia, which is associated with an excessive response to endotoxins, leading to intrahepatic inflammation and fibrosis. Clinically, the relationship between gut bacteria and NAFLD has been targeted in some randomized controlled trials of probiotics and other agents, but the results have been inconsistent. A recent randomized, placebo-controlled study explored the utility of lubiprostone, a treatment for constipation, in restoring intestinal barrier function and improving the outcomes of NAFLD patients, marking a new phase in the development of novel therapies targeting the intestinal barrier. This review summarizes recent data from studies in animal models and randomized clinical trials on the role of the gut–liver axis in NAFLD pathogenesis and progression.

## 1. Introduction

Nonalcoholic fatty liver disease (NAFLD) is the hepatic manifestation of metabolic syndrome and the leading cause of chronic liver disease in pediatric and adult populations living in industrialized countries. NAFLD encompasses steatosis and nonalcoholic steatohepatitis (NASH), and it is characterized by perivenular and lobular inflammation. Progression to fibrosis and cirrhosis are the primary complications of NAFLD [1]. Based on a recent meta-analysis, one in four people in Europe, the United States, and Asia have NAFLD [2]. The “multiple parallel hit” hypothesis may explain the pathogenesis and progression of NAFLD. Especially in recent years, there has been increasing interest in gut–liver axis dysfunction (dysbiosis, bacterial overgrowth, and changes in intestinal permeability) associated with the progression of NAFLD; therefore, gut–liver axis dysfunction is considered important as a possible alternative therapeutic target for patients who are unable to benefit from lifestyle changes, healthy eating, and promotion of physical activity [3,4]. In NASH, chronic inflammation is triggered by hepatocyte fat accumulation, followed by exposure to inflammatory cytokines, insulin resistance, oxidative stress, lipotoxicity mainly from free fatty acids (FFAs), and gut-derived endotoxins. Here, we focused on gut-derived endotoxins and reviewed the most recent data regarding the gut–liver axis and its role in the pathogenesis and progression of NAFLD. We also reviewed experimental studies in animal models and preliminary results from several randomized clinical trials (RCTs). The objectives of our review were to (1) appraise the pathophysiology of the gut–liver axis focusing on endotoxins and (2) delineate novel therapeutic perspectives via intestinal permeability.

## 2. Hepatic Inflammation and NAFLD

The inflammatory state associated with the metabolic syndrome is unique because it is not accompanied by signs of infection or autoimmunity. Furthermore, there is no major tissue damage. In addition, the dimension of inflammatory activation is not large and is often referred to as low-grade chronic inflammation. Other researchers have named this inflammatory condition metainflammation, which corresponds to metabolically induced inflammation [5], or parainflammation, which is defined as an intermediate state between basal and inflammatory states [6]. Whatever the terminology, the inflammatory processes that characterize metabolic syndrome have unique features and mechanisms that are not fully understood [7]. Metabolic syndrome, especially NASH, is caused by metainflammation. Indeed, chronic mild inflammation is an important factor in the pathogenesis of NASH [8,9].

Since the portal vein is the direct venous outflow of the intestine, the liver is continuously exposed to factors that have an intestinal origin, including bacteria and bacterial components. The liver is an important site for bacterial phagocytosis and clearance because it contains the largest population of tissue macrophages. Experimental data demonstrate that exposure of membrane components of Gram-negative bacteria and bacterial products to proinflammatory mediators such as LPS results in the activation of Kupffer cells. Kupffer cells, the resident macrophages of the liver, are the nitrogenous species that contribute to liver damage in the presence of proinflammatory cytokines and are a major source of inflammatory mediators such as ROS [10]. Through pattern recognition receptors, including Toll-like receptors (TLRs), the innate immune system recognizes conserved pathogen-associated molecular patterns (PAMPs) [11]. Although healthy livers have a low TLR mRNA level and a high resistance to TLR ligands from the constant exposure of the microflora, TLR-mediated signaling plays a major role in liver physiology and pathophysiology [12]. LPS is a potent activator of innate immune responses as it can bind to the TLR4 complex. TLR4 is expressed by Kupffer cells, hepatic stellate cells, hepatocytes, biliary epithelial cells, sinusoidal endothelial cells, and hepatic dendritic cells, which are consequently responsive to LPS [13]. There is a positive correlation among hepatic dysfunction, increased bacterial transfer, and LPS. Furthermore, in conditions with hepatic dysfunction such as cirrhosis, the clearance of LPS from circulation is reduced [13]. Cytoplasmic screening and selective recruitment of signaling adapter proteins target downstream of TLR4 signals through interactions between Toll/IL-1 receptor (TIR) domains [14,15,16]. Therefore, TLR4 activation involves the bone marrow differentiation factor 88 (MyD88), an adapter protein containing the TIR domain, or the MyD88 adapter-like factor, which is associated with the nuclear factor kappa B (NF-κB) and AP-1 transcription factor. There is substantial evidence that TLR4-mediated intracellular events exacerbate liver damage in fatty liver [17,18].

Kupffer cells produce tumor necrosis factor-α (TNF-α) and interleukin (IL)-10 in response to physiological concentrations of LPS [19,20,21]. TNF-α is an inflammatory cytokine that activates various signaling cascades, including many pathways that are described below, as important inhibitors of insulin action. TNF-α is overexpressed in obese rodents and in human adipose tissue and liver. Furthermore, it has a decreased concentration when the body is undergoing weight loss. Geoffrey et al. [22] explored the administration of a methionine–choline-deficient (MCD) diet to TNF-α and NF-κB knockout animals. Activation of TNF-α and NF-κB is essential for hepatic inflammatory recruitment in steatohepatitis. Furthermore, NF-κB activation occurs independently of TNF-α. Other studies using the MCD dietary model have reported that curcumin, which blocks oxidative stress-mediated NF-κB activation, provides protection [23], but TNF-α antiserum reduces liver injury in rats administered the MCD diet [24]. Tomita, et al. [25] demonstrated that TNF-receptor knockout mice were protected against liver fibrosis in their MCD experiments.

Alternatively, activated macrophages, including Kupffer cells, which are also referred to as M2, as opposed to the classical M1 or pro-inflammatory phenotype, represent another important pathway in resolving the inflammatory response. The coordinated program of alternative activation is primarily stimulated by the Th2 cytokines, IL-4 and IL-13, and is characterized by cell-surface expression of M2 signature genes, such as mannose receptor, arginase-1, and dectin-1 [26]. There is evidence that adiposity promotes Th1 polarization of cytokine balance, favoring congenital or classical activation of macrophages in NAFLD. Thus, in experimental and human NAFLD, the pool of hepatic natural killer T cells (NKT) is reduced, while the pool of hepatic tissue Th1 cytokines such as TNF-α, IL-12, IL-18, and interferon-γ is elevated [27,28,29,30].

## 3. Endotoxins and NAFLD

Fatty liver is caused by excessive calories due to overeating, obesity, and lack of exercise. The mechanism by which a fatty liver develops includes exposure to inflammatory cytokines, IR, oxidative stress, lipotoxicity mainly due to FFAs, and genetic predisposition. Nevertheless, exposure of the liver to endotoxins from intestinal bacteria is important.

Endotoxin levels in the blood are high in NASH patients [31]. In recent years, it was reported that serum endotoxin levels increased when volunteers were fed a high-fat Western diet, referred to as metabolic endotoxemia [32]. Gram-negative bacilli in the intestinal microflora are considered the largest source of stable endotoxins. As the number of anaerobic Gram-negative rods increases due to the deterioration of the intestinal environment, the amount of LPS, the so-called endotoxin, also increases. Although it is unlikely that all of these endotoxins enter the portal vein from the intestine due to the presence of an elaborate immune system and a functional intestinal barrier, it is still likely that some of them reach the liver and serve as a stable source of endotoxins. When enterobacteria from the intestine invade the portal vein, the first target organ is the hepatic vasculature. This suggests that Enterobacteriaceae-derived endotoxins are important in the inflammatory response that leads to the development of NASH. Endotoxins are PAMPs, which are members of a group of receptors called TLRs and nucleotide-binding oligomerization domain receptors (NLRs).

In particular, TLR4 is expressed in the plasma membrane of hepatocytes and Kupffer cells. When endotoxin stimulates TLR4, signaling molecules such as NF-kB are activated. This leads to the production of inflammatory cytokines (IL-1b and IL-18), which result in liver injury. In addition, antimicrobial therapy reduces hepatic damage in NASH. Animal studies have shown that TLR4-deficient mice did not develop NASH, thereby suggesting that intestinal bacteria play an important role in the emergence of NASH. In terms of elucidation of the mechanism of intestinal bacterial NASH development, the following are the intestinal factors: (1) disruption of the intestinal barrier function, which results in a leaky gut, (2) qualitative and quantitative dysregulation of intestinal bacteria, referred to as small intestinal bacterial overgrowth (SIBO), and (3) increased endogenous alcohol concentration [33]. On the other hand, the increased responsiveness to endotoxin is a hepatic factor (Figure 1).

## 4. Intestinal Factors Associated with Endotoxins

### 4.1. Intestinal Permeability

The concept that increased intestinal permeability and intestinal microbiota might contribute to development of some diseases was first proposed in 1890 [34]. Crosstalk between the gut and the liver may explain changes in the hepatobiliary system associated with several inflammatory and infectious bowel diseases, such as celiac disease, and infections caused by *Salmonella* and *Yersinia* [35]. Evidence of involvement of the hepatointestinal system in the development of NASH has recently emerged [36,37,38].

Obesity increases intestinal permeability through indirectly damaging the intestinal barrier [38,39,40]. An animal study showed that a high-fat diet (HFD) may increase metabolic endotoxinemia and reduce intestinal *Bifidobacteria* [41,42], which can lower intestinal LPS levels and improve mucosal barrier function [39]. Mechanisms that regulate the health of the intestinal barrier may also regulate the degree of endotoxinemia [38,39,40,43]. Another potential pathway that can be involved in the absorption of LPS in the intestine is the secretion of chylomicrons from enterocytes rather than disrupting tight junctions between cells. Studies using cell culture and animal models have suggested that endotoxins are actively secreted into the blood along with chylomicrons. Furthermore, inhibition of chylomicron synthesis inhibits endotoxin secretion [44]. These data show that gut microbiota-derived endotoxins are strongly associated with the development of NASH through impaired intestinal barriers and increased chylomicron secretion of enterocytes.

Under physiological conditions, tight junction proteins, such as zonula occludens, seal apical junctions between intestinal endothelial cells. Enteric dysbiosis disrupts these tight junctions and increases mucosal permeability, which exposes intestinal mucosal cells and the liver to potential inflammatory products. The recognition of toxic substances in the liver is mediated by pattern-recognition receptors (PRRs), a group of receptors that includes TLRs and NLRs. In a meta-analysis by Luther et al. [45], patients with NAFLD exhibited an increased intestinal permeability compared with that of healthy controls (NAFLD group: 39.1%; healthy controls: 6.8%), with an odds ratio of 5.08 (95% confidence interval (CI): 1.98–13.05). Furthermore, they reported that patients with NASH had an increased intestinal permeability compared with that of the healthy controls, with an odds ratio of 7.21 (95% CI: 2.35–22.13).

### 4.2. SIBO

Qualitative or quantitative imbalances of complex intestinal microbiota might have serious health consequences for a macroorganism, including SIBO. The discovery of a linkage between SIBO and NAFLD [46,47,48], and the observation that endotoxin triggers liver inflammation in mice with steatosis [49], prompted the formulation of this hypothesis [50]. The prevalence of SIBO was about three times that of the controls [51]. Results consistent with those demonstrating SIBO were found in 2.5% to 22% of studies investigating small sets of clinically healthy people as controls [52,53,54,55,56,57,58,59,60]. Animal studies reported that excessive multiplication of *Escherichia coli* coexisted in NASH rats [61], consistent with previous studies. This suggested that SIBO is one of many factors important in the pathogenesis of NASH, as antibacterial treatment could alleviate the severity of NASH [61]. Thus, SIBO may coexist with NASH. In addition, levels of ALT increased or decreased relative to serum levels of TNF-α [61]. This strongly supported TNF-α as an important mediator for the promotion of NASH by SIBO. Generally, endotoxemia is thought to be a link between SIBO and elevated TNF-α levels [46,62]. The association between SIBO and NAFLD and the increased endotoxemia across studies highlight the role of gut microbiota in the initiation and development of metabolic liver disease [42,52]. Lichtman et al. showed that antibiotics (metronidazole and tetracycline) reduced hepatic injury in rats with surgically induced intestinal bacterial overgrowth [48]. Drenick et al. and Vanderhoof et al. also showed that antibiotics prevented and reversed hepatic steatosis and liver injury after intestinal bypass for patients with morbid obesity [63,64]. Additionally, Bergheim et al. showed that antibiotics could reduce hepatic steatosis and endotoxinemia in a fructose-induced rodent NAFLD model [65]. These findings imply a critical role for small bowel flora, suggesting that intestinal bacterial overgrowth treatment reduces ethanol and LPS levels.

### 4.3. Ethanol-Producing Bacteria

Gut microorganisms directly cause liver damage either by means of microbe-associated molecular patterns (MAMPs) and PAMPs, such as LPS, or by means of the products of their metabolism, such as ethanol, short-chain fatty acids, and trimethylamine [66]. Proteobacteria, particularly Enterobacteriaceae, can ferment carbohydrates to ethanol. Significant correlations between the presence of ethanol-producing bacteria, blood ethanol levels, and liver inflammation have been demonstrated, and a positive correlation was found between increased abundance of Proteobacteria/Enterobacteriaceae/Escherichia and serum alcohol levels [33]. Under adequate conditions, the amount of ethanol produced can be remarkable [67]. Aside from conferring direct toxic effects to the liver, this overproduction activates hepatic ethanol metabolic pathways and increases liver oxidative stress [68]. Zhu et al. investigated gut microbiota (GM) composition and ethanol levels in the blood of NASH, obese, and healthy children [69]. The GM composition of NASH showed slight differences in the lineage, family, and genera of Proteobacteria, Enterobacteriaceae, and *Escherichia coli* compared to obese patients without liver disease. Among these microbiome changes was an increase in alcohol-producing bacteria, with significantly higher ethanol levels in NAFLD patients compared with those of obese and healthy children. In addition, an increase in ethanol concentration was particularly correlated with NASH. These results suggest that the production of ethanol by GM leads to hepatotoxicity that contributes to the onset of NAFLD and the progression to NASH [69]. Increased permeability, endogenous ethanol, and systemic endotoxin levels reflect some gut–liver axis dysfunction associated with obesity and its hepatic complications. In this regard, our group has recently demonstrated that the value of the lactulose/mannitol ratio is comparable to the grade of liver damage, significantly correlating with concentrations of ethanolemia and endotoxemia. Increased permeability was a risk factor for the development of steatosis [70].

## 5. Hepatic Factors Associated with Endotoxins

### 5.1. Enhanced Response to Endotoxin

Previous studies have shown that gut microbiota-derived endotoxins may be involved in the progression of NASH from simple fat deposition to steatohepatitis [31,38,39,40,61,62,71,72,73]. Despite these findings, the impact of increased endotoxinemia on the progression of NASH is controversial. It is still unclear whether serum endotoxin levels in NASH patients are significantly higher than those in control subjects and patients with simple fat deposition. Harte et al. reported that serum endotoxin levels were elevated in NAFLD patients compared to those in healthy controls [31]. In another human study, plasma IgG levels against endotoxin were increased in biopsy-proven human NASH patients and progressively increased with NASH grade [74]. These findings suggest a relationship between chronic endotoxin exposure and human NASH severity in which increased permeability drives endotoxemia, which in turn triggers inflammatory cytokine responses and IR [42]. However, in the study by Loguercio et al. [32,75], all NAFLD patients tested were free of endotoxinemia; however, the results were inconsistent. Currently, there is general agreement that mild portal endotoxemia can be detected in healthy subjects due to gut-derived bacterial endotoxins [40]. However, the levels of portal endotoxemia observed under healthy conditions do not usually cause liver dysfunction [76]. Furthermore, Imajo et al. reported that increased levels of leptin lead to overexpression of CD14 via activation of STAT3 signaling in Kupffer cells, resulting in a hepatic hyperinflammatory response to gut-derived low-dose bacterial endotoxin and progression from simple steatosis to steatohepatitis with associated liver inflammation and fibrosis [77]. Previous studies have also shown that a high-cholesterol diet increases the sensitivity of mice to LPS without affecting plasma levels of LPS. This further supports our hypothesis [78]. CD14 is an important regulatory factor in LPS-induced inflammation and enhances the LPS effects in Kupffer cells [79,80,81,82,83,84]. Furthermore, a previous report showed that promoter polymorphisms of CD14 are a risk factor for human NASH [85]. Therefore, increased expression of CD14 is closely related to the pathogenesis of NASH. Indeed, Imajo et al. showed that CD14 mRNA expression levels were much higher in NAFLD patients, including NAFL and NASH patients, than in control subjects [77]. Thus, hepatic CD14 may serve as an important factor in the development of NASH by enhancing hepatic inflammation against gut-derived bacterial endotoxin. We also investigated the leptin-dependent increase in hepatic CD14 expression using leptin-deficient ob/ob mice and leptin receptor-deficient db/db mice. Leptin and STAT3 signaling increased the responsiveness to gut-derived, low-dose bacterial endotoxin even in the healthy liver via an increase in CD14-positive Kupffer cells, regardless of the presence of steatosis. In humans, elevated serum leptin levels are generally associated with obesity, visceral fat accumulation, and fat deposition [86,87]. Thus, enhanced expression of hepatic CD14 by leptin may increase hepatic responsiveness to gut microbiota-derived endotoxins even at low levels, resulting in the progression from simple steatosis to NASH via STAT3 signaling. Moreover, Kessoku et al. demonstrated that resveratrol, a natural polyphenol, administration-mediated improvement of inflammation and fibrosis was due to inhibition of LPS reactivity controlled by CD14 expression in Kupffer cells. These findings suggest that resveratrol could be a candidate agent for the treatment of NASH [88].

### 5.2. Responses to Gut-Derived Ethanol

Under normal conditions, alcohol is constantly produced in the human body [88]. The intestinal microbiota is the major source of endogenous alcohol, as indicated by increased blood alcohol level after intake of alcohol-free food [89,90,91]. This endogenously produced alcohol is immediately and almost completely removed from portal blood by liver alcohol dehydrogenases (ADHs), catalases, and the microsomal ethanol-oxidizing system. When the action of ADH is inhibited, blood alcohol levels increase [89]. Production of ethanol in the gut is also reflected by the fact that the liver and the gastrointestinal tract exhibit the highest activity of ADHs [92]. Elevated breath alcohol levels are observed in obese mice. Here, the aberrant intestinal microbiota is the source for increased alcohol production, and neomycin treatment decreases alcohol concentration [93]. As patients with NASH are generally obese, and their liver histology is the same as that observed in alcoholic liver disease, it was hypothesized that NASH patients also have elevated blood alcohol levels [93]. The alcohol hypothesis of NASH could also explain the observation of increased gut permeability [40] and, possibly, elevated serum lipopolysaccharide levels in NASH patients [94], since alcohol is known to increase gut permeability [95]. The first evidence in support of this hypothesis was that the gene expression associated with all three major pathways for ethanol catabolism in the NASH liver is significantly elevated [70]. Recently, elevated blood ethanol concentration was observed in patients with NAFLD [96]. The observation of Volynets et al. [96] provides a link between blood alcohol and NAFLD. Zhu et al. further clarified that the blood ethanol concentration of obese patients without NASH is not elevated; however, obese patients with NASH exhibit significantly elevated blood ethanol levels [33].

## 6. NASH/NAFLD Treatment Focused on the Gut

### 6.1. Probiotics and Prebiotics

Probiotics are living microorganisms that regulate the intestinal flora and promote physical health. The most common probiotics on the market are *Lactobacillus*, *Streptococci*, and *Bifidobacteria*. Prebiotics are indigestible carbohydrates that stimulate the growth and activity of beneficial bacteria, especially *Lactobacillus* and *Bifidobacteria*. Lactulose is an example of a prebiotic that increases the number of *Bifidobacteria*, whereas fructooligosaccharides, including oligofructose, and inulin increase the abundance of *Lactobacillus rhamnosus* G and *Bifidobacteria lactis* Bb12 [97].

Influencing the intestinal flora using probiotics in mice with fatty liver reduced intestinal inflammation and improved epithelial barrier function [41,98], indicating that probiotics can be a new treatment for NAFLD in humans. Regarding clinical trials, Loguercio et al. [99,100] showed that probiotics reduced liver damage and improved liver function in NAFLD patients. However, subsequent pediatric meta-analyses emphasized that probiotic treatment of patients with NAFLD and nonalcoholic steatohepatitis is not recommended due to the lack of robust evidence from RCTs. Recently, a double-blind RCT showed that administration of *Lactobacillus bulgaricus* and *Streptococcus thermophiles* reduced hepatic transaminase levels in biopsy-proven adults with NAFLD [101,102]. Another double-blind RCT study showed that obese children with NAFLD treated with *Lactobacillus* GG for 8 weeks demonstrated significant decreases in ALT levels [103]. Prebiotics are indigestible food ingredients that have beneficial effects on the host by selectively stimulating the growth of selected gut microbiota and altering metabolic activity. The health benefits of prebiotics are transmitted to the host by three broad mechanisms, which include improved glucose regulation, altered lipid metabolism, and selective regulation of the gut microbiota [104]. Studies involving obese rats demonstrated that prebiotic fibers improved or normalized the intestinal microbiota dysbiosis by increasing the amount of *Firmicutes* and decreasing the amount of *Bacteroidetes phylae* [105].

These promising preliminary results strongly demonstrate that probiotics and prebiotics have great potential for the prevention and treatment of NASH. However, as pointed out in a recent meta-analysis, further clinical research is needed to clarify this unique, yet cost-effective, strategy [101]. Currently, substantial experimental evidence shows the beneficial effects of probiotics and prebiotics, which may be useful for designing future clinical trials.

### 6.2. Fecal Microbiota Transplantation

In addition to probiotic supplementation, transplantation of fecal microbiota has been shown to reduce HFD-induced steatohepatitis through modulation of the GM. In fact, fecal microbiota transplantation from lean donors to NASH patients is under investigation in humans (NCT02469272). Despite its routine use, it is difficult to consider in current clinical practice, but the results of the study help to better define the etiological mechanisms of NAFLD [4].

### 6.3. Anti-LPS Immunoglobulins

A recent promising NAFLD treatment is oral supplementation with IMM-124e, an IgG-rich bovine colostrum extract from cows immunized against LPS, which improved liver fat, insulin sensitivity [106], and immune-mediated colitis in animal models [107]. One study showed improved glycemic control in a small human trial [108]. Benefits seem to be due to reduction of liver exposure to GM LPS and consequent Kupffer cells activation.

### 6.4. Vitamin B6

The gut microbiota plays a chief role in vitamin production. Micronutrients, especially vitamins, play a crucial role in several metabolic reactions. Vitamin B6 is one of the vital micronutrients. Vitamin B6 (VitB6) is a generic name that includes six vitamers: pyridoxine (PN), pyridoxal (PL), pyridoxamine (PM), and their respective phosphate esters pyridoxine 5′-phosphate (PNP), pyridoxal 5′-phosphate (PLP), and pyridoxamine 5′-phosphate (PMP) [3]. VitB6 intake and hepatic steatosis are negatively correlated, and patients with NAFLD have diets low in VitB6 compared with healthy individuals [12,13]. In addition, patients with NAFLD have low levels of plasma VitB6, and VitB6 administration ameliorated hepatic lipid accumulation in mice model [14]. There are no reports on the efficacy of VitB6 administration in patients with NAFLD examined in clinical trials, but a study by Kobayashi and Kessoku et al. [109] was the first to demonstrate that 12 weeks of VitB6 administration ameliorates liver lipid deposition in patients with NAFLD. Although the mechanism by which VitB6 ameliorates hepatic lipid accumulation has not been fully elucidated, an experimental study reported that one potential mechanism could be linked to homocysteine (Hcy) catabolism. PLP, a physiologically active form of VitB6, functions as a coenzyme of cysteine-b-synthase (CBS) and cystathionine-g-lyase (CGL) [19]. Because CBS and CGL contribute to Hcy catabolism, inadequate amounts of VitB6 result in the accumulation of Hcy [110]; Hcy causes protein misfolding in the endoplasmic reticulum (ER), which leads to the ER stress response. In turn, ER stress induces the activation of the transcription factor sterol response element binding protein 1c, causing de novo lipogenesis [111]. Thus, based on this mechanism, VitB6 deficiency is thought to induce hepatic lipid accumulation. Glutathione is synthesized in cells from glutamic acid, cysteine, and glycine. Cysteine and glycine are generated from methionine and serine, respectively, and glutamic acid is synthesized from α-ketoglutarate, a metabolite of glucose. Glutathione has a long history in the treatment of chronic liver disease through intravenous injection; however, Honda et al. [112] demonstrated a therapeutic effect of glutathione through oral administration in patients with NAFLD. The primary outcome of this study was a change in ALT levels.

### 6.5. Vitamin D

Vitamin D deficiency also seems to play a role in NAFLD, but the underlying mechanisms are not well understood. Recently, the possible involvement of vitamin D in the dysregulation of the gut–liver axis is gradually becoming apparent. Indeed, optimal vitamin D levels are essential for maintaining the integrity of intestinal permeability through the upregulation of tight junction components and mucosal proteoglycans in the ileal epithelium, and through defensins and their converting enzymes (matrix metalloproteinase 7—MMP7) by certain intestinal mucosal paneth cells. The presence of vitamin D deficiency in the mouse HFD model exacerbates leaky gut, enterotoxemia, endotoxemia, systemic inflammation, and consequently IR and hepatic lipidosis [113]. Thus, supplementation with vitamin D has been recommended [114].

### 6.6. Constipation Drug Lubiprostone (LUB)

LUB is a bicyclic fatty acid derived from a prostone metabolite of prostaglandin E1. It is a type 2 chloride channel activator that causes an efflux of chloride into the gastrointestinal lumen, which ultimately promotes intestinal fluid secretion [115]. Thus, LUB is normally used for treating both chronic idiopathic constipation and irritable bowel syndrome with constipation. However, there are potentially additional actions of LUB on the intestinal mucosa. For example, some experimental studies reported that LUB prevented small bowel injury induced by nonsteroidal anti-inflammatory drugs (NSAIDs) in rats [116], ameliorated increases in intestinal permeability induced by a Western diet in an atherosclerotic mouse model [117], and maintained intestinal tight junction barrier function by activating the chloride channel in Caco-2 cell line [118]. Furthermore, some clinical trials demonstrated that LUB dramatically improved the intestinal permeability induced by NSAIDs in healthy volunteers [119]. Recently, a parallel three-arm, double-blind RCT was conducted using LUB as a new therapeutic target for the treatment of NAFLD, focused on intestinal permeability. A total of 150 Japanese NAFLD patients with constipation were treated with a placebo, 12 μg, or 24 μg of LUB for 12 weeks. In the LUB groups, the urinary lactulose/mannitol ratio, an index of intestinal permeability, was improved. Liver enzymes, liver fat, and blood endotoxin levels were significantly improved [120]. In particular, in the group of patients with improved intestinal permeability, a marked decrease in liver enzymes, liver fat content, and blood endotoxin concentration was observed (Figure 2). Drugs targeting intestinal permeability may be a promising new therapeutic approach to NAFLD.

## 7. Conclusions

The pathogenesis and progression of NAFLD/NASH are associated with an increased susceptibility to endotoxins, qualitative and quantitative abnormalities of intestinal bacteria, and an increase in intestinal permeability. Advances obtained in the understanding of the role of the gut–liver axis in NAFLD pathogenesis and the encouraging results already obtained by gut microbiota modulation via probiotic supplementation provide a promising and safe innovative mode of therapy. However, other extensive and long-term studies are needed to better define the best probiotic strains, their doses, timing, and duration of supplementation therapy. This will serve to individualize probiotic therapy with a patient-tailored approach for modulating intestinal permeability, endotoxemia, and treating liver disease Future studies should explore the pathogenesis of NAFLD, in particular, the mechanism involving intestinal permeability, to facilitate the development of novel therapeutic approaches targeting the gut–liver axis.

## Figures and Tables

**Figure 1 ijms-22-08161-f001:**
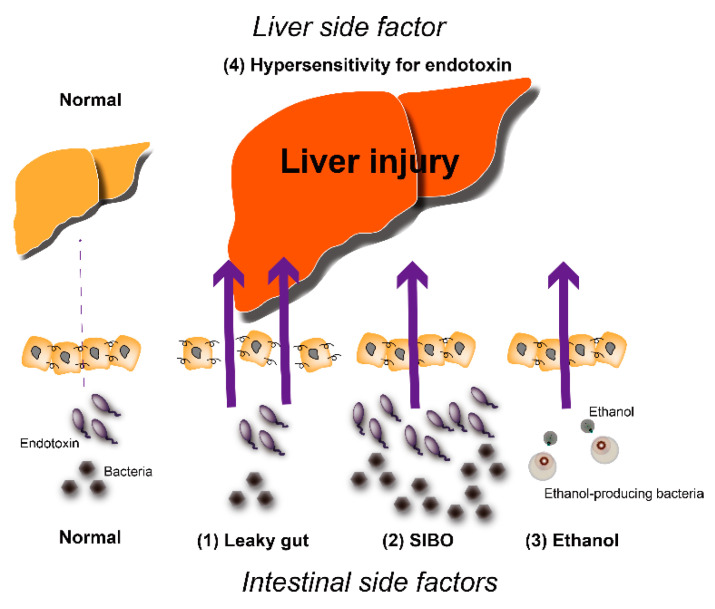
Mechanisms of NAFLD progression promoted by intestinal and hepatic side factors. SIBO; small intestinal bacterial overgrowth.

**Figure 2 ijms-22-08161-f002:**
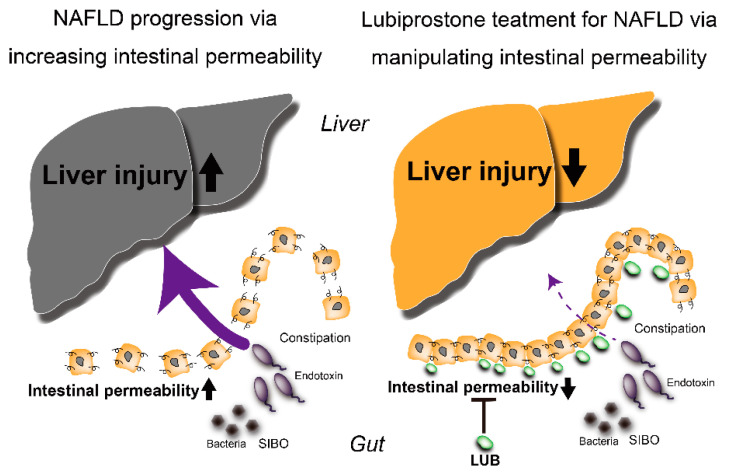
Therapeutic effects of lubiprostone on NAFLD via targeting the intestinal barrier. LUB, lubiprostone; NAFLD, nonalcoholic fatty liver disease; SIBO, small intestinal bacterial overgrowth.

## Data Availability

Not applicable.

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
