# Peer review of "The Role of Leaky Gut in Nonalcoholic Fatty Liver Disease: A Novel Therapeutic Target"

_ijms, 2021, doi:10.3390/ijms22158161_

Round 1

Reviewer 1 Report

This review may of interest for many readers, because there is a lot of curiosity about this topic. 

The paper is clear and the topics well organized, however:

  • On page 6 line 7 the sentence must be rewritten
  • On page 7 line 1 the sentence must be rewritten

Reviewer 2 Report

Studies on the interorganic communications between the gut and the liver emerge at the focal point of metabolism research. Although the number of related review articles is exponentially increasing in the past decade, there remains an enormous need to often pack up the latest discoveries due to the expanding scale of the original research on this particular topic. This manuscript aims to summarise recent findings of the gut-liver axis in the context of NAFLD, from the bench to the bedside. If doing well, it will definitely arouse broad interest to readers.     

However, the writing of the manuscript should be improved:

  1. authors should improve the quality of Figure 1. There are a few white lines alongside Endotoxin and Bac for unknown reasons.
  2. “(SIBO)” appears twice, in the last para of 3 and in the subheading of 4.3.
  3. Section 4.3 should be expanded.
  4. As the authors stated, “Increased permeability, endogenous ethanol, and systemic endotoxin levels reflect some gut-liver axis dysfunction associated with obesity and its hepatic complications.” However, the permeability is currently discussed in both 4.1 and 4.2, while endotoxin in 4.1 and 4.3.). Section 4 is better to be re-organised into three sub-sections (permeability, endogenous ethanol) accordingly, with their implications in NAFLD at the end of each sub-section.
  5. Why is there only 5.1 in section 5? The responses to gut-derived ethanol should be included as 5.2.
  6. The “target” in the subheading of section 6 should be “targets”.
  7. Does sub-sections 6.1-6.4 and 6.6 belong to “Therapeutic target for NASH/NAFLD via leaky gut?” These sub-sections do not mention any relevance to the leaky gut.
  8. The Introduction highlighted that the content of this manuscript is focused on animal data and clinical trials. Can authors re-write the paper clearly with these two sections by re-organising the content? After reading the current version, readers could not access the desired information straightforwardly.
